# A Note on the Hierarchy of Quantum Measurement Incompatibilities

**DOI:** 10.3390/e22020161

**Published:** 2020-01-30

**Authors:** Bao-Zhi Sun, Zhi-Xi Wang, Xianqing Li-Jost, Shao-Ming Fei

**Affiliations:** 1School of Mathematical Sciences, Qufu Normal University, Qufu 273165, China; 2School of Mathematical Sciences, Capital Normal University, Beijing 100048, China; wangzhx@cnu.edu.cn; 3Max Planck Institute for Mathematics in the Sciences, 04103 Leipzig, Germany; xianqing.li-jost@mis.mpg.de

**Keywords:** quntum measurements, hierarchy of incompatibilities, semidefinite program, 03.67.-a, 32.80.Qk

## Abstract

The quantum measurement incompatibility is a distinctive feature of quantum mechanics. We investigate the incompatibility of a set of general measurements and classify the incompatibility by the hierarchy of compatibilities of its subsets. By using the approach of adding noises to measurement operators, we present a complete classification of the incompatibility of a given measurement assemblage with *n* members. Detailed examples are given for the incompatibility of unbiased qubit measurements based on a semidefinite program.

## 1. Introduction

The incompatible measurements are one of the striking features in quantum physics, and can be traced back to Heisenberg’s uncertainty principle [1] and wave-particle duality [2,3]. There are observables like position and momentum, that are impossible to be exactly measured simultaneously, unless some amount of noise is added [4]. The existence of incompatible measurements [5,6,7] also implies the no-cloning theorem [8,9] and Einstein-Podolsky-Rosen (EPR) steering [10,11,12].

The incompatibility of quantum measurements is also known to be a powerful tool in many branches of quantum information theory [13,14,15,16,17,18,19]. As a resource for quantum information processing [20], quantum incompatibility has been the object of intense research [21,22,23,24,25,26,27,28,29,30,31,32,33,34,35,36,37,38]. In [39] Bell-like inequalities have been presented by using some partial compatible measurements.

Based on the investigation on the uncertainty relations of two measurements [15,16,17], the authors in [40] found that a triple measurement uncertainty relation deduced from uncertainty relations for two measurements is usually not tight. There exist genuinely incompatible triple measurements such that they are pairwise jointly measurable, as in the case of genuine tripartite entanglement or genuine nonlocal correlations. Based on statistical distance, Qin, et al. [41] formulated state-independent tight uncertainty relations, satisfied by three measurements, in terms of their triple joint measurability. Another way to quantify the joint measurability of a set of measurements is to add noise to the measurement operators, and numerically calculate the noise threshold for the measurements to be jointly measurable [21,25,26].

In this paper, similar to the quantum multipartite entanglement or non-locality, we classify the measurement incompatibility for a given set of measurements, and present a hierarchy of quantum measurement incompatibilities. We then study the transition between different types of incompatible measurements by using the semidefinite programm (SDP) [34]. We present a criterion to judge the incompatibility of a given multiple measurements. Detailed examples are given for unbiased qubit measurements.

## 2. Measurement Incompatibility Classification and Quantification

A general positive operator valued measure (POVM) is given by a collection of positive-semidefinite operators summing up to an identity. We consider a POVM with measurement operators ${\left\{A_{a}\right\}}_{a}$, $\{A_{a}\geq 0$, $\sum_{a}A_{a}=I\}$, where I stands for the identity operator. Given a state ρ, the probability of measurement outcome *a* is $p\left(a\right)=\mathrm{tr}A_{a}\rho$.

Given two POVMs, $\left\{A_{a}\right\}$ and $\left\{B_{b}\right\}$, $\left\{A_{a}\right\}$ is called a coarsening of $\left\{B_{b}\right\}$, or equivalently, $\left\{B_{b}\right\}$ is called a refinement of $\left\{A_{a}\right\}$, if the former can be derived from the latter by data processing, $A_{a}=\sum_{b}p\left(a|b\right)B_{b}$, for every *a*, where p(a|b)≥0, and ${\left\{p\left(a|b\right)\right\}}_{a}$ is a probability distribution, for every *b* [35]. For a set of *n* POVMs, i.e., a POVM assemblage $A={\left\{{\left\{A_{a|x}\right\}}_{a}\right\}}_{x=1}^{n}$, we say that the POVM assemblage is compatible (or jointly measurable) if and only if all the POVMs in the assemblage possess a common refinement. This common refinement is called the parent POVM [34]. Otherwise, A is incompatible.

The incompatibility of a POVM assemblage with *n* members implies that there does not exist a common refinement for all of the *n* measurements. However, some of the measurements of the assemblage may have a common refinement, that is, there might be some subassemblages which are jointly measurable. Obviously, if a subset of a POVM assemblage is incompatible, the whole assemblage is incompatible. However, the converse is not true. Therefore, it is of significance to characterize the measurement incompatibility of a POVM assemblage in a finer way, as in the separability classification in quantum entanglement [42,43]. To give a uniform description of the measurement incompatibility of a POVM assemblage with *n* measurements, we have the following classifications:

Given a POVM assemblage $A={\left\{{\left\{A_{a|x}\right\}}_{a}\right\}}_{x=1}^{n}$ and k≤n, we have
(1)A is (n,k)-compatible, if all *k*-member subsets of A are compatible;(2)A is (n,k)-incompatible, if at least one *k*-member subset of A is incompatible;(3)A is (n,k)-strong incompatible, if all *k*-member subsets of A are incompatible;(4)A is (n,k+1)-genuinely incompatible, if A is (n,k)-compatible, and (n,k+1)-incompatible;(5)A is (n,k+1)-genuinely strong incompatible, if A is (n,k)-compatible, and (n,k+1)-strong incompatible.

From above classification, we have the following conclusions:
**Proposition** **1.***Given a set of POVMs $A={\left\{{\left\{A_{a|x}\right\}}_{a}\right\}}_{x=1}^{n}$, we have**(1)* A is compatible, if and only if, it is (n,n)-compatible;*(2)* A is (n,k+1)-compatible implies that it is (n,k)-compatible;*(3)* (n,k)-strong incompatible is a special case of (n,k)-incompatible;*(4)* (n,k)-incompatible implies (n,k+1)-incompatible but not (n,k+1)-genuinely incompatible;*(5)* (n,k+1)-genuinely incompatible means that (n,k+1)-incompatible and (n,k)-compatible;*(6)* (n,2)-(strong) incompatible is equivalent to (n,2)-genuinely (strong) incompatible.

For a given POVM assemblage A with *n* members, an interesting problem is to judge if it is incompatible. If A is incompatible, we need to determine which kind of incompatibility it is. If A is (n,k)-incompatible or (n,k)-strong incompatible, it would become (n,k)-, (n,k+1)- or even (n,n)-compatible by adding more and more white noise.

Adding noise to a POVM assemblage $A={\left\{{\left\{A_{a|x}\right\}}_{a}\right\}}_{x}$ is to mix each measurement operator in A with white noise, so as to get a new set of POVMs $A^{\eta }$,
(1)$$A_{a|x}^{\eta }=\eta A_{a|x}+\left(1-\eta \right)\mathrm{tr}A_{a|x}\frac{I}{d}.$$

As ${\left\{{\left\{\mathrm{tr}A_{a|x}\frac{I}{d}\right\}}_{a}\right\}}_{x}$ is a compatible assemblage, $A^{\eta }$ will eventually become jointly measurable for sufficient small η. The critical parameter $\eta^{*}$, at which the transition from incompatible to compatible occurs, is called the noise robustness of A, which is a meaningful incompatibility quantifier [21,25,26]. The bigger $\eta^{*}$, the weaker incompatibility of A. A is compatible if and only if $\eta^{*}=1$. It is generally formidably difficult to obtain the accurate value of $\eta^{*}$. The estimation of the upper or lower bound of $\eta^{*}$ is an interesting problem. In [34] Designolle et al. gave an expression of $\eta^{*}$ by using SDP (Equation 2) and its strong dual (Equation 3),
(2)$$\begin{array}{ccc} \eta^{*}= & \max & \eta \\ & s.t. & \{A_{a|x}^{\eta }\}\;\mathrm{compatible}\\ & & 0\leq \eta \leq 1, \end{array}$$
(3)$$\begin{array}{ccc} \eta^{*}= & \min_{{\left\{X_{a|x}\right\}}_{a,x}} & 1+\mathrm{tr}\sum_{a,x}X_{a|x}A_{a|x}\\ & s.t. & 1+\mathrm{tr}\sum_{a,x}X_{a|x}A_{a|x}\geq \frac{1}{d}\sum_{a,x}\mathrm{tr}X_{a|x}\mathrm{tr}A_{a|x},\\ & & \sum_{x}X_{j_{x}|x}\geq 0,\;\;\;\forall j_{i},\;\;i=1,2,\cdots ,n, \end{array}$$
where any ${\left\{X_{a|x}\right\}}_{a,x}$ that satisfies the constraints in (Equation 3) can give rise to an upper bound of $\eta^{*}$.

The critical parameter $\eta^{*}$ can be also used to characterize the transition from general (n,n)-incompatible to (n,n)-compatible, as well as from (n,k)-incompatible to (n,k)-compatible, or (n,k)-strong incompatible to (n,k)-incompatible. We denote $\eta_{\left(n,k\right)}^{*}$ the critical parameter at which the transition from (n,k)-incompatible to (n,k)-compatible occurs. The $\eta^{*}$ given in (Equation 2) and (Equation 3) is simply $\eta_{\left(n,n\right)}^{*}$. If A is (n,k)-compatible, then $\eta_{\left(n,k\right)}^{*}=1$. Otherwise, $\eta_{\left(n,k\right)}^{*}<1$. Since (n,k)-incompatible implies the (n,k+1)-incompatible, we have $\eta_{\left(n,k\right)}^{*}\geq \eta_{\left(n,k+1\right)}^{*}$.

Denote [n]={1,2,⋯,n}. Let $\alpha_{k}$ represent an arbitrary subset of [n] with *k* numbers. Given any *k*-member subset of A, $A_{\alpha_{k}}=\left\{{\left\{A_{a|x}\right\}}_{a}|x\in \alpha_{k}\right\}$. If $A_{\alpha_{k}}$ is (k,k)-incompatible, by using the same SDP (Equation 2) and SDP-dual (Equation 3) procedure, we can get the critical number $\eta_{\alpha_{k}}^{*}$, at which the transition from (k,k)-incompatible to (k,k)-compatible occurs for $A_{\alpha_{k}}^{\eta }$,
(4)$$\begin{array}{ccc} \eta_{\alpha_{k}}^{*}= & \max & \eta \\ & s.t. & {\left\{A_{a|x}^{\eta }\right\}}_{x\in \alpha_{k}}\;\mathrm{compatible}\\ & & 0\leq \eta \leq 1, \end{array}$$
(5)$$\begin{array}{ccc} \eta_{\alpha_{k}}^{*}= & \min & 1+\mathrm{tr}\sum_{x\in \alpha_{k}}\sum_{a}X_{a|x}A_{a|x}\\ & s.t. & 1+\mathrm{tr}\sum_{x\in \alpha_{k}}\sum_{a}X_{a|x}A_{a|x}\geq \frac{1}{d}\sum_{x\in \alpha_{k}}\sum_{a}\mathrm{tr}X_{a|x}\mathrm{tr}A_{a|x},\\ & & \sum_{x\in \alpha_{k}}X_{j_{x}|x}\geq 0,\;\;\;\forall j_{i},\;\;i\in \alpha_{k}. \end{array}$$

For any subset with *k* members $A_{\alpha_{k}}$ of A, $A_{\alpha_{k}}^{\eta }$ is (k,k)-compatible if $\eta \leq \eta_{\alpha_{k}}^{*}$, and (k,k)-incompatible if $\eta >\eta_{\alpha_{k}}^{*}$. Set
(6)$$\eta_{\left(n,k\right)}^{min}=\min_{\alpha_{k}}\eta_{\alpha_{k}}^{*},\;\;\;\;\;\;\eta_{\left(n,k\right)}^{max}=\max_{\alpha_{k}}\eta_{\alpha_{k}}^{*}.$$

We have $\eta_{\left(n,k\right)}^{*}=\eta_{\left(n,k\right)}^{min}$. $\eta_{\left(n,k\right)}^{min}$ and $\eta_{\left(n,k\right)}^{max}$ can identify all kinds of incompatibilities of a POVM assemblage. If $\eta_{\left(n,k\right)}^{min}=\eta_{\left(n,k\right)}^{max}=1$, namely, $A_{\alpha_{k}}$ is compatible for all $\alpha_{k}$, then A is (n,k)-compatible. If $\eta_{\left(n,k\right)}^{min}<1$ and $\eta_{\left(n,k\right)}^{max}=1$, A is (n,k)-incompatible but not strong incompatible. If $\eta_{\left(n,k\right)}^{max}<1$, we conclude that A is (n,k)-strong incompatible. If $\eta_{\left(n,k+1\right)}^{min}<1$ and $\eta_{\left(n,k\right)}^{min}=1$, then A is (n,k+1)-genuinely incompatible. If $\eta_{\left(n,k+1\right)}^{max}<1$ and $\eta_{\left(n,k\right)}^{min}=1$, then A is (n,k+1)-genuinely strong incompatible. Therefore, we have the following theorem.

**Theorem** **1.**
*The numbers $\eta_{\left(n,k\right)}^{min}$ and $\eta_{\left(n,k\right)}^{max}$, defined in (Equation 6), can classify all kinds of incompatibility of a POVM assemblage $A={\left\{{\left\{A_{a|x}\right\}}_{a}\right\}}_{x=1}^{n}$ for k=2,3,⋯,n.*


From another point of view, in the case of $\eta_{\left(n,k\right)}^{min}<\eta_{\left(n,k\right)}^{max}\leq 1$, $A^{\eta }$ is (n,k)-strong incompatible for $\eta >\eta_{\left(n,k\right)}^{max}$, because every $A_{\alpha_{k}}^{\eta }$ is incompatible. $A^{\eta }$ is (n,k)-compatible for $\eta \leq \eta_{\left(n,k\right)}^{min}$, and (n,k)-incompatible but not strong incompatible for $\eta_{\left(n,k\right)}^{min}<\eta \leq \eta_{\left(n,k\right)}^{max}$. In this sense, we can say that $\eta_{\left(n,k\right)}^{min}$ is the critical parameter for the transition from (n,k)-compatible to general (n,k)-incompatible. $\eta_{\left(n,k\right)}^{max}$ is the one for the transition to (n,k)-strong incompatible.

It is clear that $\eta_{\left(n,k\right)}^{min}\geq \eta_{\left(n,k+1\right)}^{min}$ and $\eta_{\left(n,k\right)}^{max}\geq \eta_{\left(n,k+1\right)}^{max}$. Nevertheless, the general relation between $\eta_{\left(n,k\right)}^{min}$ and $\eta_{\left(n,k+1\right)}^{max}$ is not clear. If $\eta_{\left(n,k+1\right)}^{max}<\eta_{\left(n,k\right)}^{min}$, then $A^{\eta }$ is (n,k+1)-genuinely strong incompatible for $\eta_{\left(n,k+1\right)}^{max}<\eta \leq \eta_{\left(n,k\right)}^{min}$. But if $\eta_{\left(n,k+1\right)}^{max}<\eta_{\left(n,k\right)}^{min}$, $A^{\eta }$ is not (n,k+1)-genuinely strong incompatible for all η.

By considering “maximally incompatible” measurements, the authors in [44,45,46,47] discussed the projective measurements on mutually unbiased bases (MUBs), the bases regarded as “maximally noncommutative” and “complementary” [44]. MUBs play a central role in quantum information processing [48], and have been used in a wide range of applications [49,50,51,52,53,54].

In the following, we give examples of different kinds of incompatibility by using projective measurements on mutually unbiased bases in qubit systems. Given three mutually unbiased bases ${\left\{\{|\psi_{a|x}\rangle \right\}}_{a=1}^{2}{\}}_{x=1}^{3}$,
$$|\langle \psi_{a|x}|\psi_{b|y}\rangle {|}^{2}=\frac{1}{2},\;\;x\neq y,\;\;a,b=1,2,$$
and
$$\langle \psi_{a|x}|\psi_{b|x}\rangle =\delta_{ab},\;\;x=1,2,3,\;\;a,b=1,2.$$

We consider the projective measurements given by these bases, $A={\left\{\{A_{a|x}=|\psi_{a|x}\rangle {\langle \psi_{a|x}|\}}_{a}\right\}}_{x=1}^{3}$. Correspondingly, there exist three unit real 3-dimensional vectors ${\vec{m}}_{x}$, such that
$$\{A_{a|x}=|\psi_{a|x}\rangle {\langle \psi_{a|x}|\}}_{a=1,2}=\left\{A_{\pm |x}=\frac{I\pm {\vec{m}}_{x}\cdot \vec{\sigma }}{2}\right\},\;\;\;x=1,2,3.$$

The mutual unbiases of $\{|\psi_{a|x}\rangle {\}}_{a}$, x=1,2,3, gives that ${\vec{m}}_{1}$, ${\vec{m}}_{2}$, and ${\vec{m}}_{3}$ are mutually orthogonal. Adding noise to $A_{a|x}$, we find $A^{\eta }={\left\{A_{\pm |x}^{\eta }=\eta A_{\pm |x}+\left(1-\eta \right)\frac{I}{2}\right\}}_{x=1}^{3}$, namely,
$$A^{\eta }={\left\{A_{\pm |x}^{\eta }=\frac{I\pm \eta {\vec{m}}_{x}\cdot \vec{\sigma }}{2}\right\}}_{x=1}^{3}.$$

Using the results in [31,34,36], we have critical parameters $\eta_{\left(3,2\right)}^{*}=\eta_{\left(3,2\right)}^{min}=\eta_{\left(3,2\right)}^{max}=\frac{1}{\sqrt{2}},$
$\eta_{\left(3,3\right)}^{*}=\frac{1}{\sqrt{3}}$. Hence, if $\eta \leq \frac{1}{\sqrt{3}}$, then $A^{\eta }$ is (3,3)-compatible. If $\frac{1}{\sqrt{3}}<\eta \leq \frac{1}{\sqrt{2}}$, then $A^{\eta }$ is (3,2)-compatible, but (3,3)-incompatible, i.e., (3,3)-genuinely incompatible. If $\eta >\frac{1}{\sqrt{2}}$, then $A^{\eta }$ is (3,2)-strong incompatible.

On the other hand, we can add different white noise to different POVMs in A. We have new POVM assemblage as ${\left\{A_{\pm |x}^{\eta_{x}}\right\}}_{x=1}^{3}$, $\eta_{x}\in \left[0,1\right]$. Then $A_{\pm |x}^{\eta_{x}}$ and $A_{\pm |y}^{\eta_{y}}$, with 1≤x<y≤3, are compatible if $\eta_{x}^{2}+\eta_{y}^{2}\leq 1$ [29]. If $\eta_{1}=\eta_{2}\equiv \eta >\frac{1}{\sqrt{2}}$ and $\eta_{3}^{2}+\eta^{2}\leq 1$, then $\{A_{\pm |1}^{\eta_{1}}$, $A_{\pm |2}^{\eta_{2}}\}$ are incompatible, but $A_{\pm |x}^{\eta_{x}}$ and $A_{\pm |3}^{\eta_{3}}$ are compatible (x=1,2). Hence $\left\{A_{\pm |1}^{\eta_{1}},A_{\pm |2}^{\eta_{2}},A_{\pm |3}^{\eta_{3}}\right\}$ is an example for (3,2)-incompatible but not (3,2)-strong incompatible.

## 3. Conclusions

We have investigated the measurement incompatibility of a general measurement assemblage. We have classified such quantum into (n,k)-compatible, (n,k)-incompatible and (n,k)-strong incompatible. By using the approach of mixing with noises, detailed examples are presented. Incompatibility and compatibility of measurements play profound roles not only in the fundamental research of quantum physics, but also in quantum information processing, raging from uncertainty relations to the detection of Bell nonlocality and device-independent certification of entanglement. Finer characterization of the incompatibility of measurement assemblages can give rise to better applications. Our results may highlight further researches on jointly measurability and applications of incompatible measurements in quantum information processing.

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
