# Peer review of "A Note on the Hierarchy of Quantum Measurement Incompatibilities"

_entropy, 2020, doi:10.3390/e22020161_

Round 1

Reviewer 1 Report

In the manuscript "A note on hierarchy of quantum measurement  incompatibilities", the authors present a classification for a given  measurement assemblage with n members.   The definitions "compatible, incompatible, strong incompatible, genuinely  incompatible and genuinely strong incompatible" are quite natural. The critical  parameter eta(n,k)^* is used to characterize the noise-robustness of the  assemblage. It quantifies how much noise need to be added such that an  (n,k)-incompatible becomes (n,k)-compatible.   Finally, the authors consider an example with three mutually unbiased bases in  qubit systems. They compute the critical parameters eta(n,k)^* and classify the  compatibility of the assemblage as the noise levels eta varies.   I feel that the manuscript is an obvious generalization of existing definitions  and results. I don't find it very interesting, but it's fine as an article. I  recommend it be published.

Author Response

We thank the referee very much for the kind recommendation.

In addition, we have corrected minor typos.

Reviewer 2 Report

The authors discuss the problem of multi-measurement incompatibility writing down a hierarchy of criteria. They also discuss the degree of incompatibility considering the amount of noise necessary to experience the transition from compatible to incompatible sets of measurements.

The bibliography could be in my opinion better completed citing for instance recent papers such as P. Coles et al., RMP 89, 015002 (2017); F. Hirsch et al., PRA 97, 012129 (2018); G. Styliaris and P. Zanardi, Phys. Rev. Lett. 123, 070401 (2019). As a very minor comment, in line 25, where there should be the word “set” there is “st”.

The manuscript is certainly interesting, the results are clear and well presented. So, I will recommend it for publication in Entropy.

Author Response

We thank the referee very much for appreciating our results.

We have replaced "st" in  line 25 with "set", and added three references ([4],[5],[6]):

P. J. Coles, M. Berta, M. Tomamichel, S. Wehner, Rev. Mod. Phys. {\bf 89}, 015002 (2017).

F. Hirsch, M. T. Quintino, N. Brunner, Phys. Rev. A {\bf 97}, 012129 (2018).

G. Styliaris and P. Zanardi, Phys. Rev. Lett. {\bf 123}, 070401 (2019).

as advised by the referee.

Reviewer 3 Report

In the paper the authors study the incompatibility in the set generalized measurements given by positive operator valued measures. The set of POVMs is compatible iff all POVMs have common refinement. In the case of sets of n measurements incompatibility means that there does not exist a common refinement of all n measurements, but there might be some subsets which are jointly measurable. In the paper the authors present complete classification of the hierarchy of incompatibilities in the set of n measurements.  In my opinion the manuscript can be accepted for publications, but the authors should correct all misprints. I noticed the following:

lines 19 and 20 line 25 references 20 and 42

Author Response

1) lines 19 and 20

We have replaced

"the authors in [40] found that a triple measurement uncertainty relation, which shows that a triple measurement uncertainty relation deduced from uncertainty relations for two measurements is usually not tight."

with

"the authors in [40] found that a triple measurement uncertainty relation deduced from uncertainty relations for two measurements is usually not tight."

2) line 25

corrected already in the last version.

3) references 20 and 42

We have corrected the spelling of the authors.